# MONITRS: Multimodal Observations of Natural Incidents Through Remote Sensing

**Shreelekha Revankar**[1]*, **Utkarsh Mall**[2], **Cheng Perng Phoo**[1], **Kavita Bala**[1], **Bharath Hariharan**[1]

[1]Cornell University [2]Columbia University

## Abstract

Natural disasters cause devastating damage to communities and infrastructure every year. Effective disaster response is hampered by the difficulty of accessing affected areas during and after events. Remote sensing has allowed us to monitor natural disasters in a remote way. More recently there have been advances in computer vision and deep learning that help automate satellite imagery analysis, However, they remain limited by their narrow focus on specific disaster types, reliance on manual expert interpretation, and lack of datasets with sufficient temporal granularity or natural language annotations for tracking disaster progression. We present MONITRS, a novel multimodal dataset of more than 10,000 FEMA disaster events with temporal satellite imagery and natural language annotations from news articles, accompanied by geotagged locations, and question-answer pairs. We demonstrate that fine-tuning existing MLLMs on our dataset yields significant performance improvements for disaster monitoring tasks, establishing a new benchmark for machine learning-assisted disaster response systems.

## 1 Introduction

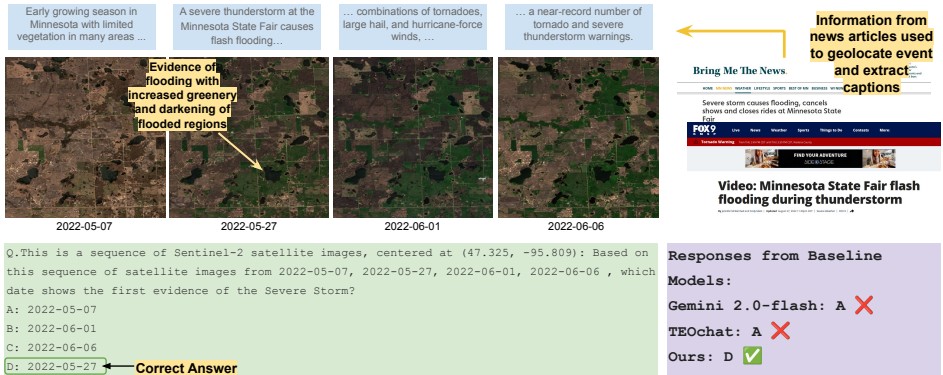

Figure 1: Using news articles, we extract exact locations of disaster events and corresponding captions for event timelines. Our MONITRS dataset enables precise disaster monitoring, as shown in this Minnesota severe storm sequence. The May 27th image shows evidence of flooding with increased vegetation and darker water-saturated regions. Models finetuned with MONITRS correctly identify the temporal onset of the storm while baseline models fail to detect the initial evidence.

---

*Corresponding Email: revankar@cs.cornell.edu

39th Conference on Neural Information Processing Systems (NeurIPS 2025) Track on Datasets and Benchmarks.

Natural disasters cause significant damage to infrastructure, homes, and communities, resulting in loss of life and billions of dollars in economic costs annually. Effective disaster response depends on understanding what events are occurring, where they are taking place, and how they progress over time [6]. However, affected regions are often inaccessible or dangerous to access during and after disasters.

A promising solution is automatic analysis of satellite imagery, enabling non-invasive coverage of disaster zones [3]. However, natural disasters pose unique challenges for such analysis: they are characterized by rapid change in a short period of time, and understanding this rapid temporal evolution is critical for disaster management. Unfortunately, much of the recent literature on recognizing concepts in satellite imagery focuses on static concepts like land-use and is not equipped to analyze rapid change events like natural disasters. Approaches that do detect change often do not allow for semantic interpretation [39] or do not provide fine-grained temporal understanding [4, 13, 14]. The few approaches that have been proposed specifically for natural disasters either focus on specific disaster types with specialized models [37, 2] or require substantial manual interpretation by domain experts [8].

A key challenge in building recognition models for disaster understanding is the lack of annotated datasets. However, building such a dataset is difficult: natural disasters are by definition rare, and straightforward sampling of remote-sensing imagery is unlikely to chance upon these events. Even if we were to get remote sensing imagery from natural calamities, they are not annotated with the kinds of concepts we may want recognized. For instance, many of the available annotations for satellite imagery revolve around land-use, which is why existing approaches can recognize when buildings are built, but not where wildfire scarring has occurred. This lack of annotations cannot be resolved easily through manual annotations because remote sensing imagery is an unfamiliar domain for most lay annotators.

In this paper, we address this data challenge by presenting MONITRS (**M**ultimodal **O**bservations of **N**atural **I**ncidents **T**hrough **R**emote **S**ensing) — a first-of-its-kind dataset of remote-sensing imagery of natural disasters annotated with natural language descriptions. Our key insight is to pair public records of natural disasters in the US maintained by the Federal Emergency Management Agency (FEMA) with *news articles* covering these events and containing detailed natural language descriptions. We propose a novel data curation pipeline that combines these sources to produce a unified resource for disaster monitoring research and application development.

MONITRS consists of approximately 10,000 disaster events documented by FEMA, paired with:

- Temporal sequences of geolocated satellite imagery capturing each event's progression,
- Natural language annotations derived from news articles describing the events,
- Precise geotagged locations marking areas of interest within each event, and finally
- Question-answer pairs designed to train and evaluate multimodal language models

Unlike existing disaster monitoring datasets that focus on single disaster types or limited temporal windows, MONITRS captures the complete lifecycle of diverse disaster events, from initial impact through recovery phases.

Using our dataset, we demonstrate that existing remote-sensing multimodal LLMs (MLLMs) are indeed unable to understand the progression of natural disasters. We find that existing models are particularly bad at temporal grounding and event classification for natural disasters. To address these limitations, we fine-tune existing MLLMs on our dataset and demonstrate improved performance in the domain of disaster response.

Our work addresses a significant gap in disaster monitoring resources and lays the groundwork for more effective, machine learning-assisted disaster response systems that combine the geographic comprehensiveness of satellite imagery with the accessibility of natural language interfaces.

## 2 Related Works

### 2.1 Event Monitoring using Earth Observation Data

Many ML methods have been used to model temporal sequences of earth observation data. Particularly in disaster monitoring, automated methods for change detection can help in planning disaster

relief, assessing damage extent, and monitoring recovery. These approaches typically analyze pairs or sequences of images capturing the same location over time to identify changes that indicate disasters [33, 39, 27].

Disaster monitoring presents unique challenges compared to general change detection tasks, as changes can be sudden and dramatic and require models that can distinguish between normal changes (for example, seasonal changes) and disaster-induced ones [30, 21, 23]. Prior works have explored various approaches for disaster-specific applications, including building damage assessment [2], flood extent mapping [37], wildfire tracking [38], and post-disaster recovery monitoring [36]. However, most existing approaches are designed for specific disaster types or short temporal windows. This limits the types of disasters that any one system can monitor [34].

While change detection techniques have made significant progress in identifying visual differences between temporal imagery, they typically lack natural language understanding capabilities [21, 24]. Some specialized models can identify and distinguish certain events, but they can only process limited time sequences, making them insufficient for comprehensive disaster monitoring that requires tracking changes over extended periods [4, 14, 13].

## 2.2   Vision-Language Models for Earth Observation Data

Efforts to develop VLMs for EO data have been rapidly increasing. These methods commonly use different single-image EO datasets and convert them to instruction-following tasks, then fine-tune a LLaVA-like model on the dataset [15, 14].

Recent works have introduced novel image-caption datasets for training remote sensing foundation models, pairing aerial and satellite imagery with captions generated using landmarks or utilizing public web images with the text filtered for the remote sensing domain [31, 22, 20]. These approaches have demonstrated state-of-the-art generalization performance in zero-shot retrieval.

Most existing VLMs for Earth Observation are designed to handle single image inputs, limiting their use for many real-world tasks that require temporal reasoning, particularly for phenomena like natural disasters that evolve over time [16].

Several recent works have developed VLMs that can engage in conversation about videos, demonstrating the potential for temporal reasoning in multimodal models [17, 40]. Approaches such as TEOChat [14] have shown that video-language models can be adapted to handle temporal sequences of earth observation data, performing a wide variety of spatial and temporal reasoning tasks. However, these models are constrained by the lack of temporal granularity in existing training datasets for remote sensing events. This limitation prevents tracking the full progression of natural disasters.

## 2.3   Multimodal Datasets for Remote Sensing Events

Existing multimodal datasets for remote sensing typically focus on a limited set of tasks or specific disaster types [19, 42]. Various change detection datasets focused on building change [12, 2], land cover changes, or land use changes [42]. While several works have designed self-supervised approaches to leverage temporal sequences of earth observation data [39, 23, 21], few have developed comprehensive datasets that combine satellite imagery, geospatial information, and textual annotations derived from real-world sources like news articles.

The lack of large-scale, diverse datasets that include multiple disaster types, temporal scales, and annotations, presents a significant bottleneck for developing general-purpose models for disaster monitoring and response. Our work addresses this gap by creating a comprehensive dataset covering approximately 10,000 disaster events from FEMA, incorporating geolocated satellite imagery throughout the duration of events, natural language annotations from news articles, geotagged locations relevant to the events, and question-answer pairs for training multimodal language models.

## 3   MONITRS

Effective monitoring of natural disasters requires us to understand certain details about the disaster, such as where it is occurring, when it began, and how it affects the infrastructure and communities

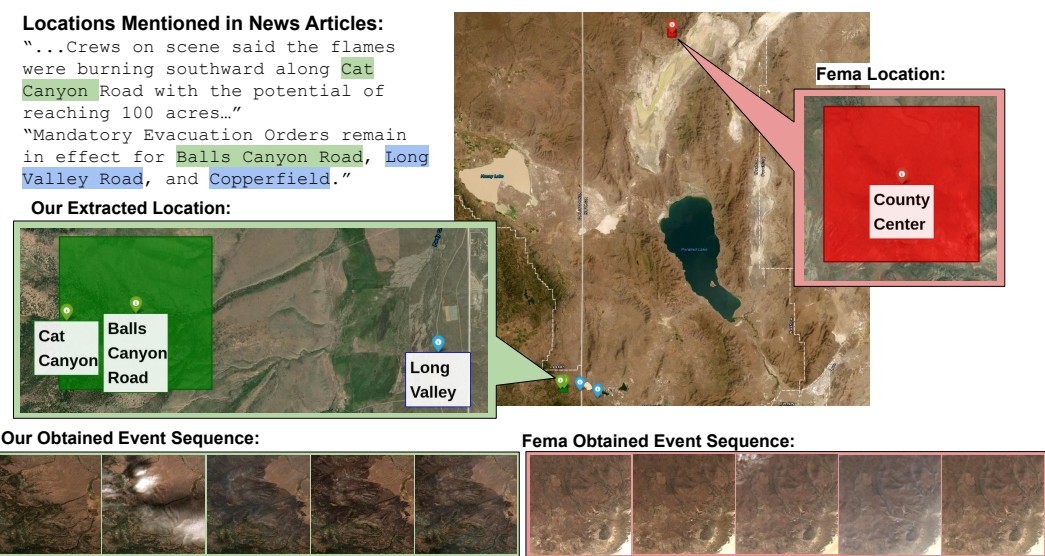

Figure 2: We demonstrate the use of geocoded news articles used to capture a better understanding of an events exact location. Here we visualize the result of our pipeline for the Loyalton Fire that took place in 2020, over the border of two neighboring states (California and Nevada). The FEMA provided coordinates for any event are the center of the county in which the event is located, however this does not necessarily provide the best coverage of the event, especially in cases like this where the disaster spans multiple counties, or in cases where the county is so large that the center coordinate is not near to the event location. Our sequence captures the progression of the fires by maintaining close distance to locations named in the news articles.

in its path. We aim to automate this process via satellite imagery so that we can perform effective monitoring over large areas in a non-invasive, less labor intensive way.

Recent works have demonstrated that large multimodal language models can act as powerful tools for understanding events [14, 17]. However, current datasets do not capture the necessary details to train such a model to act as a sufficient tool for the task at hand. We create a novel natural disaster dataset that captures the required information.

## 3.1 MONITRS Construction

The first challenge we need to address is the relative rarity of natural disasters. As such, simply sampling remote sensing imagery is unlikely to yield enough samples for these events. Instead, we begin with FEMA's Disaster Declarations Areas [7], which includes a list of all federally declared disasters. This helps us define the types of disasters we include in our scope. Since we want to acquire the relevant satellite imagery that tracks each event, we only keep events that have enough information to spatio-temporally localize the event, namely, county, state, event name, and start and end dates. Events that do not have this information are discarded.

While FEMA keeps some information of the disasters, they do not keep detailed descriptions of their extent. For example, while the records contain the county where the disaster occurred, the true locations of the disaster and its effects can be far from the exact centers of these counties. This poses a challenge in acquiring the right remote-sensing imagery that captures the full extent of the event. In addition, the FEMA database does not include any annotations or descriptions of the evolution of the event, which would be needed to train capable remote-sensing multimodal LLMs.

**News articles for events:** We find that a better way to locate the full extent of these events is to leverage news articles written about the disaster. These articles provide detailed descriptions that capture which specific regions were affected, when and how. This not only allows us to geolocate the event correctly, but also provides us with natural language descriptions that describe the evolution of the event in detail.

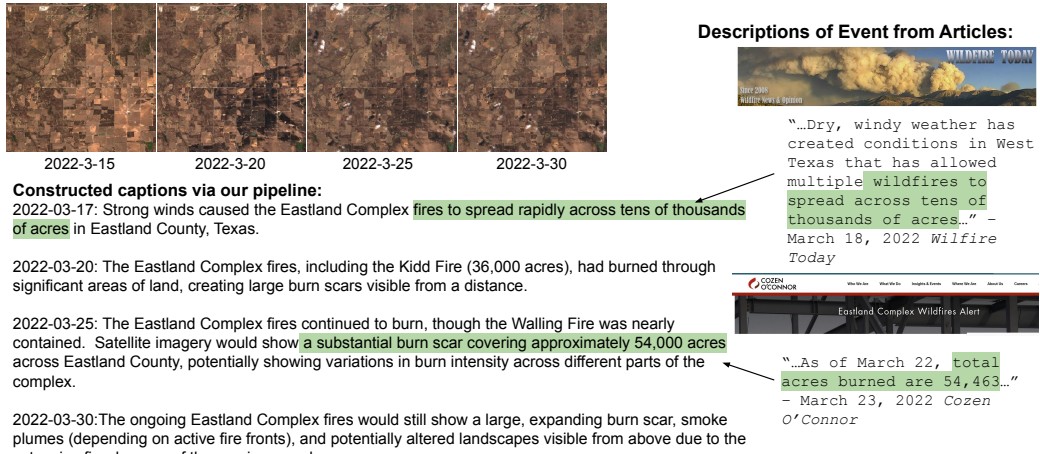

Figure 3: We illustrate the captions generated through our dataset construction pipeline. After geolocating the news articles, we prompt an LLM to retrieve captions using the articles' contents for a list of dates using the text alone. This ensures we are captioning the imagery independently of what may be visible. We see that our process accurately describes the wildfire even in Eastland, Texas.

To find relevant news articles, we construct search queries using our filtered list of FEMA events. The queries are comprised of the event name, county, state, and start date. For each event, we collect news articles or reports. To reduce the chance of accidentally including irrelevant information, we select the first five results returned by the search query, using the Google Search API [10].

From these articles, we first ascertain the exact location and geographical extent of the natural disaster being reported on. We begin by parsing through the articles using LLMs, specifically the freely available Gemini 2.0-flash model. We ask the model to retrieve all of the proper nouns of locations mentioned in the articles. For example this includes specific highways, or town names. We create a union of all the locations mentioned across the articles and retrieve their geocoded location (latitudinal and longitudinal position) using the Geocoding API [25]. This gives us a more complete representation of the extent of the event.

**Acquiring satellite images:** With these locations at hand, we select the square patch (of fixed size) that includes the maximum number of proper noun locations mentioned across all articles. This square patch forms the basis for acquiring satellite imagery. As a source of satellite images, we use RGB bands of Sentinel-2 imagery, which is publicly available [5]. Sentinel-2 imagery has a ground sampling distance of 10m per pixel and a re-visit rate of 5 days on average. The size of the square patch is $5.12 \times 5.12 km^2$, which corresponds to a 512x512 pixel image. With this region we download all available satellite images for the duration of the natural disaster as reported by FEMA, including a 10 day buffer before and after the event to ensure we capture its entirety.

**Acquiring natural language descriptions:** The final step is to produce natural language descriptions of the event. We wish to produce descriptions for the temporal evolution of the event. To this end, we make note of all of the dates that comprise the natural disaster event. We then prompt Gemini with these dates and with the text of all the news articles for the event (which includes dates as well), and ask it to describe what visible events have occurred by each date. This is done using the article content and dates alone.

Ultimately, through this process, for a set of natural disaster events we have, (a) the approximated locations of the events, (b) satellite imagery that covers the event, (c) a list of geolocated proper nouns that are affected or associated with the event, (d) detailed descriptions of the event through time captured using (e) news articles reporting on the event. The five components make up MONITRS, and can be used to support several downstream tasks.

Next, we use this dataset to create a VQA datasets to benchmark and finetune large multimodal language models for answering questions about events from satellite imagery.

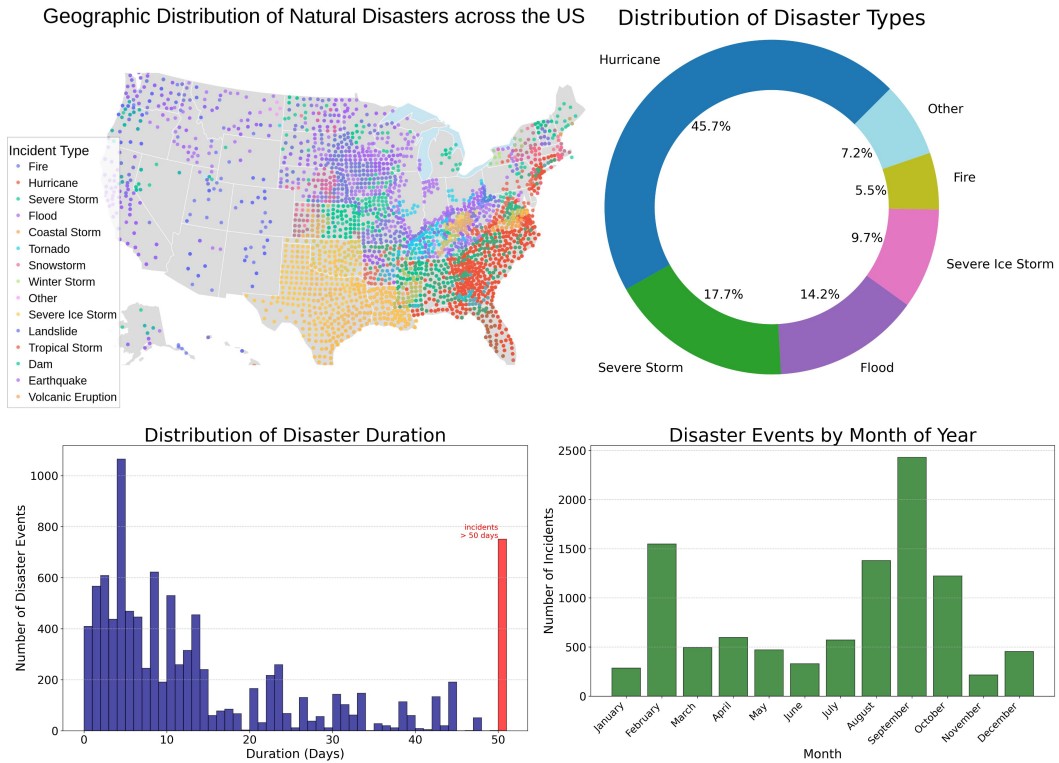

Figure 4: Our dataset represents the wide variety of natural disasters recorded by FEMA.

## 3.2 Dataset Statistics

Our dataset contains 9,996 disaster incidents collected from FEMA records. We visualize statistics about the dataset in Figure 4. Hurricanes and severe storms constitute the majority of events, with strong seasonal patterns peaking in September. Geographic distribution centers primarily in coastal and hurricane-prone regions, with the states of Louisiana, Texas, and Florida experiencing the highest incident counts. On average there are 4.13 images per event, representing on average 18.14 days.

# 4 MONITRS-QA

With MONITRS, we have sufficient information to construct a visual question-answering dataset for natural disasters. We utilize two formats of question-answer datasets for different purposes. The first being multiple-choice QA datasets, so that correct answers can be confirmed easily for quantitative results. The second being open-ended QA datasets, which allows for more detailed and descriptive responses.

We develop these datasets using two approaches. The first is templated question and answers, where we standardize questions with slots for event-specific information. Using a template allows us to evaluate model performance for specific kinds of reasoning. The second is generated question and answers, where we employ large language models to create diverse, event specific questions with linguistic variety.

**Templated questions:** The types of reasoning covered in our templated questions include *event classification*, *temporal grounding*, and *location grounding*:

*Event Classification* questions ask the model to categorize the event.

*Temporal Grounding* questions ask when the event began and when it ended.

*Location Grounding* questions focus on where the disaster is taking place, and the affected infrastructure.

| Category | Question Type | Description | Example |
|---|---|---|---|
| Templated | Event Classification | Identifying which disaster is occurring | What type of event is shown in these satellite images?
A: [EVENT_TYPE]
B: [EVENT_TYPE]
C: [EVENT_TYPE]
D: [EVENT_TYPE] |
| Templated | Temporal Grounding | Determining when disasters begin and end | Based on this sequence of satellite images from [DATES], which date shows the first evidence of the [EVENT_TYPE]? |
| Templated | Location Grounding | Identifying where disasters occur and affected infrastructure | What happened at [LOCATION] before [DATE]? |
| Generated | Event-specific MCQ | Multiple choice questions with event-specific details | Analyzing the progression of the wildfire, what appears to be the primary factor influencing its spread?
A: Strong prevailing winds pushing the fire eastward.
B: The presence of a significant amount of dry brush and easily combustible vegetation.
C: Proximity to a major water source, significantly hindering fire spread.
D: Planned burns implemented by local fire departments effectively slowing the blaze. |
| Generated | Event-specific Free-response | Questions about specific events | What were the conditions that led to the rapid spread of wildfires in Kansas, Texas, and Oklahoma? |

Table 1: Categorization of disaster-related questions in our dataset.

Our multiple choice benchmarks are balanced, with roughly the same probability for each option to be the correct answer.

**Generated questions:** For the generated question-answer datasets, we prompt LLMs to create questions that are event specific, allowing for a more diverse variety of questions that pertain more specifically to the events in question.

**Train/test splits:** We split the dataset by event to prevent location/temporal overlap. The train split contains 44,308 QA pairs, while the test set contains 10,196 QA pairs.

## 5 Experiments

**Experimental Setup** For our baseline evaluation, we include the following models:

- VideoLLaVA 7b [17]: A video-language model that has been adapted for temporal reasoning tasks.
- GeoChat [15]: A remote sensing specific video-language model, designed for single-image analysis and cannot accept temporal sequences. For this reason, we query the model using the first image in our sequences.
- TEOchat 7b [14]: A recent multimodal model specifically designed for temporal earth observation data, which should theoretically be well-suited for our task.
- Gemini 2.0-flash [9]: A lightweight state-of-the-art closed-source multimodal model.
- Gemini 2.0-pro [9]: A state-of-the-art closed-source multimodal model with demonstrated capabilities on remote sensing tasks.
- GPT-4.1 [28]: A state-of-the-art closed-source multimodal model that has demonstrated strong performance on various vision-language tasks.

We finetune TEOChat on our MONITRS-QA training set using LoRA (r=32, $\alpha$=64) with a learning rate of 2e-5, batch size 1 with 8 gradient accumulation steps (effective batch size 8), cosine learning

Table 2: Multiple Choice Event Classification & Grounding

| Method | Event Classification | Temporal Grounding | Location Grounding |
|---|---|---|---|
| Videollava [17] | 49.72% | 11.11% | 17.11% |
| GeoChat [15] | 28.18% | 26.5% | **76.80%** |
| TEOchat [14] | 48.88% | 15.15% | 15.50% |
| Gemini 2.0-flash [9] | 50.07% | 18.02% | 13.74% |
| Gemini 2.0-pro [9] | 72.06% | 14.01% | 33.81% |
| GPT 4.1 [28] | 39.12% | 21.43% | 21.63% |
| Ours (1/5 MONITRS-QA) | 88.69% | 70.72% | 23.25% |
| Ours (full MONITRS-QA) | **91.66%** | **76.05%** | 31.34% |

rate scheduler with 0.03 warmup ratio, and 8-bit quantization. Training was performed on 4 A6000 GPUs for 1 epoch. We report results for both the full training set and a reduced set (1/5 size) to assess data efficiency. Training on the reduced set took approximately 3 hours per epoch.

**Metrics** For the multiple choice question-answer datasets we report overall accuracy and perform McNemar's statistical test [26] to assess the significance of performance differences between models and validate observed improvements in MCQ tasks. For open-ended answers, we use established metrics for question-answering: BLEU [29], ROUGE-L [18], and METEOR [1], which measure n-gram overlap, longest common subsequence and semantic similarity respectively. Additionally we analyze answers using LLMs as judges, as described in Zheng et. al [41]. In general we ask Gemini 2.0-flash to score the factual accuracy, completeness, specificity, use of visual evidence, and the answer overall. We include the exact prompts in the appendix.

# 6 Results

We discuss quantitative results on MONITRS-QA in the main paper, while providing additional qualitative examples and visualizations of model predictions in the appendix.

## 6.1 Multiple Choice Event Classification and Grounding

**Current state-of-the-art:** Overall, we found baseline models struggle to answer questions related to natural disasters. For event classification, baseline performances hover around ~50%, except Gemini 2.0-pro [9] which achieves 72.06%. Performance drops even lower for temporal (11-26%) and location (13-17%) grounding, with the notable exception of GeoChat [15] achieving 76.80%.

**Results after finetuning on MONITRS-QA:** Given the poor and inconsistent performance of current state-of-the-art, we finetune TEOchat [14], using both our full MONITRS-QA training dataset as well as a reduced training set (approximately 1/5th), for 1 epoch.

As shown in Table 2, our finetuned model significantly outperforms the baselines on most multiple-choice task types. For event classification, our model achieves 91.66% accuracy on the full dataset (88.69% on 1/5 data). The gap widens further for temporal grounding, where our model achieves 76.05% accuracy on the full dataset (70.72% on 1/5 data). For location grounding, our model achieves 31.34% accuracy on the full dataset (23.25% on 1/5 data), showing improvements over most baselines though still trailing GeoChat's 76.80%.

We conducted McNemar's test [26] to assess the statistical significance of performance differences between models. Our finetuned model demonstrated statistically significant improvements over all baselines (p < 0.001). Specifically, our model correctly answered 296 questions that TEOChat missed for event classification (while TEOChat, the model specialized in temporal satellite events only correctly answered 11 questions our model missed).

**Task-Specific Challenges:** We hypothesize that the gap between results in temporal grounding and event classification may be due to the idea that some events can be classified from a single image alone, but that temporal grounding which requires looking at the entire sequence, is not being learned.

Table 3: Generated VQA

| Method | Multiple-Choice Accuracy | Open-Ended | | | | | |
|---|---|---|---|---|---|---|---|
| | | BLEU-1 | BLEU-2 | BLEU-3 | BLEU-4 | METEOR | ROUGE-L |
| Videollava [17] | 36.65% | 0.3447 | 0.2814 | 0.2490 | 0.2221 | 0.4739 | 0.3965 |
| TEOchat [14] | 36.99% | 0.3439 | 0.2805 | 0.2483 | 0.2216 | 0.4736 | 0.3951 |
| Gemini 2.0-flash [9] | 28.13% | 0.2050 | 0.1398 | 0.1123 | 0.0920 | 0.3478 | 0.2419 |
| Ours (1/5 MONITRS-QA) | 52.18% | 0.4046 | 0.3351 | 0.2969 | 0.2667 | 0.4912 | 0.4275 |

Table 4: Generated VQA – LLM Evaluation

| Method | Open-Ended | | | | | |
|---|---|---|---|---|---|---|
| | Factual Accuracy | Completeness | Specificity | Visual Evidence | Uncertainty Handling | Overall |
| Videollava [17] | 3.41 | 3.46 | 3.53 | 2.27 | 4.26 | 3.08 |
| TEOchat [14] | 3.39 | 3.45 | 3.52 | 2.28 | 4.31 | 3.08 |
| Gemini 2.0-flash [9] | 2.44 | 2.10 | 2.04 | 2.00 | 4.15 | 2.13 |
| Ours (1/5 MONITRS-QA) | 3.84 | 3.54 | 3.72 | 2.50 | 4.29 | 3.08 |

With limited finetuning, the improvement for event classification and temporal grounding is both substantial and statistically significant ($p < 0.01$ to $p < 0.001$). This suggests that models are capable of learning to identify natural disasters, but have not quite learned to pick up on the gradual changes that are needed to differentiate types of events.

Location grounding remains challenging almost all models, but even then our finetuned model maintained statistically significant improvements over baselines ($p < 0.01$ to $p < 0.001$).

Overall these results demonstrate that we have effectively created a challenging enough benchmark that even prominent MLLMs have significant room for improvement.

### 6.2 General Disaster Response VQA

From Table 3, all models showed lower overall accuracy. Our fine-tuned model maintained significant advantages (52.18% versus 28-37% for baselines, $p < 0.001$), but the performance gap slightly narrowed compared to templated tasks. Our model correctly answered over 1000 questions that each baseline missed, while failing on only 362-431 questions where baselines succeeded.

The results from the LLM-based evaluation in Table 4, suggest that fine-tuning on MONITRS improves the model's ability to connect language with visual features regarding natural disasters.

## 7 Discussion

Overall, our results demonstrate that MONITRS addresses a critical gap in disaster monitoring capabilities, with baseline models struggling on natural disaster tasks and our fine-tuned models showing substantial improvements.

We find that the location positioning task is especially difficult for some models, however our results demonstrate that this is a valid task that sufficiently trained models should be able to perform. Notably, GeoChat achieves exceptional performance on location grounding (76.80%), which supports our hypothesis that models specifically trained on geospatial relationships can excel at spatial localization tasks. This improved performance is likely because GeoChat had a significant portion of its training data relating to the relationship between latitude and longitude and pixel correspondence [15].

To clarify the task: we give the models the center coordinates of the image as well as the pixel resolution, and ask it to deduce the location of a concept/feature within the image in pixel coordinates. The understanding of pixel correspondence to latitude and longitude is non trivial, as the distance covered by 1 unit longitude or latitude is different at different locations around the globe.

We found that multiframe models that accept sequences of images actually perform worse than single image models like GeoChat for tasks such as location grounding. However, this multi-frame architecture is still necessary to classify or understand the progression of temporal events.

We also see a performance discrepancy between Gemini 2.0-pro and GPT-4.1 with Gemini substantially outperforming GPT on event classification tasks. We hypothesize that Gemini has likely been

trained with labeled satellite imagery [35]. This demonstrates that we have effectively created a challenging enough benchmark that even prominent MLLMs have significant room for improvement.

With these results we find that MONITRS fills a gap by aligning language descriptions with visual evidence at specific temporal stages. The significant improvement after fine-tuning shows existing architectures can learn disaster recognition and temporal progression in satellite imagery when sufficiently trained with specialized data.

**Future Applications.** The MONITRS dataset offers potential value beyond the immediate disaster classification and description tasks we've explored. Some promising directions include:

- Representation Learning: The aligned multimodal nature of MONITRS is well-suited for learning representations for change events, potentially creating embeddings that capture the semantic meaning of various disaster stages even without accompanying images.
- Architectural Innovations: Future work could explore new architectural components like date/time embeddings that explicitly encode temporal information in models, improving their ability to reason about disaster events through time.
- Beyond Disasters: While this dataset currently contains data regarding natural disasters, there is room for generalization as the geolocating of events is done using articles. Our methodology could potentially be extended to other domains with other events that are documented in news and lack sufficient visual annotations.

**Limitations.** While we see a number of applications and models that could benefit from our dataset, there are several limitations worth discussing.

Our dataset relies on FEMA records, which only cover U.S. disasters, limiting generalization to global disaster events that may have different visual signatures. Global datasets for geocoded natural disasters such as GDIS [32] or EM-DAT [11] are geocoded at the country/province/regional level, which is much coarser than FEMA, making it difficult to acquire the precise satellite imagery required. To our knowledge, no similar scale, validated set of global geolocated natural disasters exists in open source format. As such, our goal was to create a benchmark with available FEMA data so the community can start working on this problem.

To evaluate generalization beyond U.S. disasters, we constructed a small international test set with 18 events (detailed in Appendix D). Our fine-tuned TEOChat achieved 45.65% accuracy on international data compared to 66.35% on U.S. data (baseline [14]: 21.74% international, 26.39% U.S.), demonstrating reasonable transfer with consistent improvement over baseline in both settings, though expanded geographically diverse training data would likely improve cross-region performance.

Our imagery is sourced from Sentinel-2 [5], which has 10m per pixel resolution and approximately 5-day revisit period, which may miss critical stages in rapidly evolving disasters. However, Sentinel-2 is the highest temporal and spatial resolution satellite imagery publicly available. We include complete metadata (locations and time frames) so researchers with access to higher resolution proprietary data can expand the dataset.

While we have taken steps to ensure annotation quality, LLM-generated descriptions based on news articles may not always accurately reflect what is visible in satellite imagery. We minimize this drift using at least 5 articles per event. Human validation (detailed in Appendix C) showed most events with clear visual signatures had strong caption alignment, though resolution limitations prevent verification of fine-grained details for some disaster types.

Finally, our dataset only includes RGB satellite imagery. Additional spectral bands or synthetic aperture radar (SAR) data could provide valuable information, especially for cloud-covered regions.

# 8 Conclusion

We presented MONITRS, a novel multimodal dataset that pairs temporal satellite imagery of natural disasters with natural language descriptions derived from news articles. Our approach addresses a significant gap in existing disaster monitoring datasets by providing fine-grained temporal annotations and diverse disaster types.

**Acknowledgements**    This work was funded in part by NSF IIS 2403015. We would like to thank Arjun Devraj for his helpful discussions.

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

## A  Qualitative Results

We include qualitative examples from both MONITRS and MONITRS-QA (along with results) in Figure 5.

## B  Prompts to LLM

We use prompts to LLMs to act as language tools for two types of tasks in our work. The first being to read through and retrieve the relevant information from news articles to caption our image sequences, figures 6 and 7. The second being utilizing our captions to generate event specific question-answer pairs, figures 8 and 9.

## C  Human Validation of Caption Quality

We conducted human validation on 144 events sampled across 15 disaster types to assess caption quality. Human evaluators were asked to classify each event as: (1) clear alignment between images, captions, and sources, (2) mismatch, or (3) inconclusive where imagery was insufficient to verify caption details. Overall results showed 65.3% clear alignment between images, captions, and sources, 18.8% had mismatches, and 16.0% were inconclusive where imagery was insufficient to verify caption details. Excluding inconclusive cases, 77.7% of determinable events showed alignment, demonstrating reasonable caption quality for LLM-generated annotations.

Performance varied by disaster type, with strongest results for events with distinct visual signatures. Typhoons, tornadoes, winter storms, and dam-related events achieved 100% accuracy on clear images. Fire events showed 92.3% accuracy (12/13 clear events), coastal storms 90.0% (18/20), and floods 85.7% (6/7).

Error analysis on mismatched events revealed that snowstorms showed the highest error rates. These errors primarily stem from difficulty distinguishing white snow and ice from clouds or existing snow cover in the imagery. Hurricane events had a 35.7% mismatch rate, largely because captions describe ground-level wind damage that is not visible from satellite perspective.

The 16.0% inconclusive rate reflects a persistent challenge in validating satellite based disaster event captions. That is, captions may accurately describe events as reported in news articles, but 10m resolution imagery does not provide sufficient detail to verify specific claims. For example, descriptions of "dozens of homes destroyed" cannot be confirmed at this resolution, though large-scale burn scars or flooding extent remain visible. This does not indicate caption errors but rather highlights the resolution gap between textual descriptions from the ground level and satellite imagery. As we discuss in our limitations section 7, we provide complete location and time metadata to enable extensions with higher-resolution data sources.

## D  International Transfer Evaluation

To assess generalization beyond the United States, we curated a test set of 18 international disaster events from 8 countries across 5 continents: Greece, Chile, Spain, Ecuador, Morocco, Colombia, Libya, Japan, Canada, and Kenya. The set included 5 fires, 3 floods, and 2 earthquakes, with temporal coverage from 2023-2024.

We processed these events using our MONITRS pipeline: news article retrieval, location extraction, Sentinel-2 imagery acquisition, and caption generation. For each event, we generated templated multiple-choice questions for event classification, temporal grounding, and location grounding.

Our fine-tuned TEOChat achieved 45.65% accuracy averaged across all question types, compared to 21.74% for the baseline, TEOchat [14]. On U.S. test data, the fine-tuned model achieved 66.35% versus 26.39% by the baseline. The performance gap suggests that incorporating geographically diverse training data would improve cross-region generalization, though the current results validate that models trained on MONITRS can reasonably generalize to international disasters.

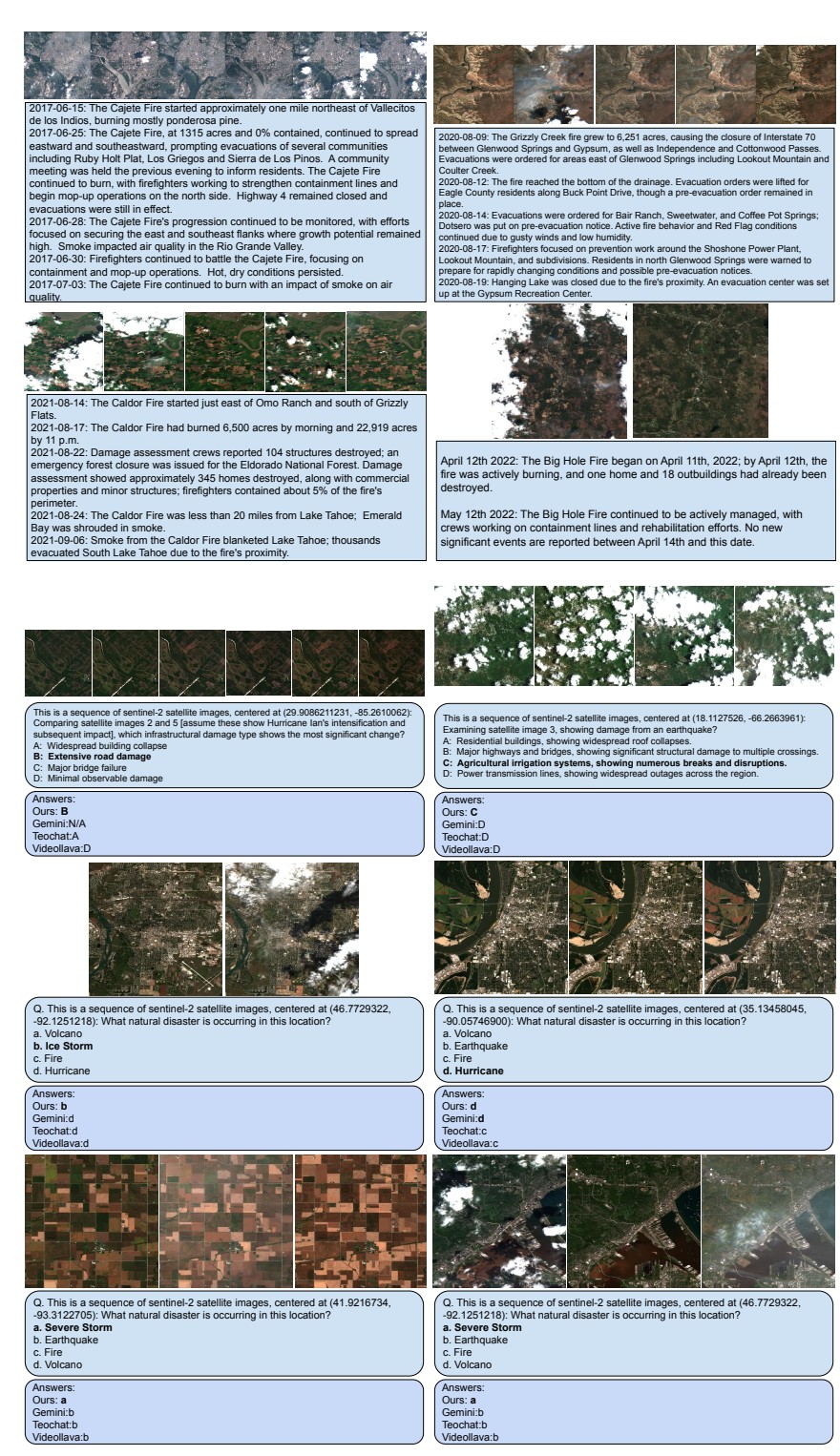

Figure 5: Qualitative examples from both MONITRS and MONITRS-QA along with their respective results.

**Task:** Extract only the event-specific geographical locations mentioned in the provided articles about natural disasters.

**Instructions:**

1. Carefully review the attached articles about natural disasters and identify ONLY proper noun locations that are directly related to where the disaster occurred or had direct impact.

2. Focus on extracting:
   - Specific sites where the event took place (cities, towns, neighborhoods)
   - Precise natural features affected (specific rivers, mountains, forests, beaches)
   - Particular infrastructure impacted (named dams, bridges, parks)
   - Exact regions directly experiencing the disaster effects

3. Present your response in a simple string list format, with each location separated by a comma.

4. If a location appears multiple times, include it only ONCE in your list.

5. If the articles contain NO specific event locations, return only the word "no" (lowercase).

6. DO NOT include:
   - Broad geographical entities not directly affected (countries, states, unless the entire entity was impacted)
   - Locations only mentioned incidentally (headquarters of responding agencies, etc.)
   - Places mentioned for context but not directly experiencing the disaster
   - General areas not specified with proper nouns

**Examples:**
For a wildfire article: `Paradise, Camp Creek Road, Butte County, Sierra Nevada foothills, Eastland County`
NOT: `California, United States, Western US`
For a hurricane article: `New Orleans, French Quarter, Lake Pontchartrain, Superdome`
NOT: `Louisiana, Gulf Coast, United States` (unless the entire state/region was directly impacted)
**Format for response when locations are found:** `Paradise, Camp Creek Road, Butte County, Sierra Nevada foothills`
**Format for response when no locations are found:** `no`
**Article Content:** `{text}`

Figure 6: Prompt given to LLM to extract proper nouns locations.

**Task:** Create a chronological timeline of observable natural disaster events from the provided news articles.

**Instructions:**

1. Review the attached news articles for information about natural disasters (earthquakes, floods, hurricanes, wildfires, volcanic eruptions, etc.).

2. For each date in the provided list, identify natural disaster events that occurred on or by that date that would be seen remotely.

3. Write a 1-2 sentence description for each date focusing specifically on the visible physical manifestations, such as:

   - Extent of flooding or inundation
   - Wildfire burn scars or active fire fronts
   - Hurricane cloud formations or aftermath flooding
   - Visible structural damage to landscapes or urban areas
   - Changes to coastlines, river courses, or terrain
   - Ash clouds, lava flows, or other volcanic features

4. If a specific date isn't explicitly mentioned in the articles, use context clues to reasonably infer when these visible changes occurred.

5. Present your response as a simple chronological list with dates followed by descriptions.

6. Emphasize the VISUAL aspects that would be detectable from above.

**Format example:**

```
June 15, 2023:  Extensive flooding covered approximately 60 square
miles of the Mississippi Delta region, with standing water clearly
visible across previously inhabited areas and farmland.
July 3, 2023:  The Caldor wildfire in California created a distinct
burn scar spanning 25 miles along the Sierra Nevada mountain range,
with active fire fronts visible on the northeastern perimeter.
```

**Article Content:** {text}

**Dates for analysis:** {dates}

Figure 7: Prompt for creating chronological timelines of visually observable natural disaster events

Given a set of statements in an order I'd like you to make 3 multiple choice questions about the events described. Make the questions diverse, covering different aspects of the events that could be answerable using satellite imagery of the event. Each question should have 4 options (A, B, C, and D) with only one correct answer.

**Statements:** \n{events}

Format your response exactly like this:

```
**Question 1:** [Your first question here] A) [First option] B) [Second
option]   C) [Third option]   D) [Fourth option]   **Correct Answer 1:**
[Correct option letter]
**Question 2:** [Your second question here] A) [First option] B) [Second
option]   C) [Third option]   D) [Fourth option]   **Correct Answer 2:**
[Correct option letter]
**Question 3:** [Your third question here] A) [First option] B) [Second
option]   C) [Third option]   D) [Fourth option]   **Correct Answer 3:**
[Correct option letter]
```

**Here are some examples of statements:** 2021-12-11: No events described in the article are visible from this date. 2021-12-15: Very strong winds in Kansas, Texas, and Oklahoma caused numerous wildfires to spread rapidly. Blowing dust severely reduced visibility, causing streetlights to turn on at midday in some areas. 2021-12-16: A large wildfire in Russell and Ellis Counties, Kansas burned approximately 365,850 acres, destroying at least 10 homes. High winds, gusting up to 100 mph, fueled the fire and other blazes across western Kansas, Oklahoma, and Texas. 2021-12-21: No events described in the article are visible from this date.

**Here are some examples of questions:**

```
**Question 1:** What natural disaster is visible in the satellite
images from mid-December 2021?   A) Hurricane B) Tornado C) Wildfire D)
Flooding **Correct Answer 1:** C
**Question 2:** Approximately how many acres were burned in Russell and
Ellis Counties, Kansas? A) 36,585 acres B) 365,850 acres C) 3,658 acres
D) 3,658,500 acres **Correct Answer 2:** B
**Question 3:** What weather condition contributed significantly to the
spread of wildfires in December 2021? A) Heavy rainfall B) Strong winds
C) Freezing temperatures D) High humidity **Correct Answer 3:** B
```

Figure 8: Prompt for generating multiple choice questions from natural disaster event statements

Given a set of statements in an order I'd like you to make 3 questions about the events described. Make the questions diverse, covering different aspects of the events that could be aided answerable using satellite imagery of the event.

**Statements:** `\n{events}`

Format your response exactly like this:

```
**Question 1:** [Your first question here]    **Answer 1:** [Your first
answer as a complete sentence]    **Question 2:** [Your second question
here]        **Answer 2:** [Your second answer as a complete sentence]
**Question 3:** [Your third question here]    **Answer 3:** [Your third
answer as a complete sentence]
```

**Here are some examples of statements:** 2021-12-11: No events described in the article are visible from this date. 2021-12-15: Very strong winds in Kansas, Texas, and Oklahoma caused numerous wildfires to spread rapidly. Blowing dust severely reduced visibility, causing streetlights to turn on at midday in some areas. 2021-12-16: A large wildfire in Russell and Ellis Counties, Kansas burned approximately 365,850 acres, destroying at least 10 homes. High winds, gusting up to 100 mph, fueled the fire and other blazes across western Kansas, Oklahoma, and Texas. 2021-12-21: No events described in the article are visible from this date. 2021-12-26: No events described in the article are visible from this date. 2021-12-31: No events described in the article are visible from this date. 2022-01-05: No events described in the article are visible from this date. 2022-01-10: No events described in the article are visible from this date. 2022-01-15: No events described in the article are visible from this date.

**Here are some examples of questions:**

```
**Question 1:** What were the conditions that led to the rapid
spread of wildfires in Kansas, Texas, and Oklahoma?        **Answer 1:**
The conditions that led to the rapid spread of wildfires in Kansas,
Texas, and Oklahoma were very strong winds, low humidity, and high
temperatures.
**Question 2:** What was the impact of the wildfires in Russell and
Ellis Counties, Kansas?    **Answer 2:** The impact of the wildfires in
Russell and Ellis Counties, Kansas was the burning of approximately
365,850 acres and the destruction of at least 10 homes.
**Question 3:** When did the wildfires in Kansas, Texas, and Oklahoma
occur?        **Answer 3:** The wildfires in Kansas, Texas, and Oklahoma
occurred on December 15, 2021.
```

Figure 9: Prompt for generating question-answer pairs from natural disaster event statements

