# OpenReview forum: "MONITRS: Multimodal Observations of Natural Incidents Through Remote Sensing"
_NeurIPS.cc/2025/Datasets_and_Benchmarks_Track — NeurIPS 2025 Datasets and Benchmarks Track spotlight_

### Official Review · Reviewer_Fv5k · 2025-06-13

**Rating:** 5
**Confidence:** 3

**Summary:**

This paper presents MONITRS, a new multimodal benchmark and dataset designed to support the development and evaluation of large multimodal language models (MLLMs) for natural disaster monitoring using temporal satellite imagery and natural language descriptions. The dataset includes over 10,000 FEMA-documented disaster events, each paired with Sentinel-2 satellite imagery, geotagged locations, news-derived captions, and question-answer pairs. The authors demonstrate that fine-tuning existing models (e.g., TEOChat) on MONITRS significantly improves performance on disaster-related tasks such as event classification, temporal grounding, and visual question answering (VQA).

**Dataset Code Accessibility:**

Partly

**Ethical Considerations:**

No, there are no or only very minor ethics concerns

**Final Justification:**

Thanks for the author's rebuttal. Most of my concerns have been addressed. I keep my rating.

**Limitations Weaknesses:**

1. The use of LLM-generated captions from news articles introduces a level of semantic abstraction that may not always align with what is visible in the satellite imagery, potentially affecting model training quality.

2. While the dataset includes geolocation data, location grounding remains a weak point across all models. The paper acknowledges this but offers limited solutions beyond suggesting possible future directions (e.g., location embeddings or segmentation masks).

3. The reported finetuning experiments are conducted on 1/5th of the full dataset due to compute constraints. While results are promising, full-scale experiments would better validate the dataset’s impact.

**Strengths Contributions:**

1. The authors introduce a multimodal dataset that combines remote sensing imagery with temporally aligned natural language descriptions derived from real-world news articles. This addresses a major gap in the current ML and EO (Earth Observation) communities.

2. MONITRS includes temporal sequences of satellite imagery, natural language captions, geolocations, and templated and free-form QA pairs, supporting a wide range of multimodal reasoning tasks.

3. The paper rigorously evaluates multiple baseline MLLMs and shows significant improvements when fine-tuning on MONITRS, especially for temporal grounding (from ~15% to 70% accuracy) and event classification (from ~50% to 88%).

4. The dataset curation pipeline is well-documented and uses LLMs to extract structured information (locations, captions) from unstructured news data. Geocoding and alignment with Sentinel-2 imagery are clearly explained.

---

> ### Author Rebuttal · Authors · 2025-07-31
>
> Thank you for your review and your positive assessment.
>
> **(1)** Regarding the issue of alignment between imagery and LLM annotations , we agree that LLMs are susceptible to biases in the input text, which is why we do not use a single article to create captions. We mitigate this potential bias and reduce noise by utilizing the information from multiple (5) relevant articles (section 3.1, page 5, lines 142-143) so that our captions are not biased by a single news article.
>
> **(2)** Yes, the location positioning task is difficult for some models, but our **new findings support that this is a valid task that sufficiently trained models should be able to perform.** During this rebuttal period, we have expanded our evaluation to include additional prominent MLLMs (Gemini 2.0 Pro, GPT-4.1). We also evaluate GeoChat [1], a single image remote sensing vlm, (using only a single image from the original sequence). Notably, GeoChat achieves exceptional performance on location grounding (76.80%), which supports our hypothesis that models specifically trained on geospatial relationships can excel at spatial localization tasks. '
>
> To clarify, the task is the following: We give the models the center coordinates of the image as well as the pixel resolution, and ask it to deduce the location of a concept/feature within the image in pixel coordinates. In the future as models utilize location encodings or additional modalities of information that encode geospatial coordinates, we should see improved performance in this task. We find that multiframe or models that accept sequences of images actually perform worse than single image models like GeoChat for tasks such as location grounding. However, this multi-frame architecture is still necessary to classify or understand the progression of temporal events.
>
> **(3)** For the scale of experimentation, we agree that a full-dataset training experiment would further strengthen our claims. For the rebuttal, we have conducted training with the full dataset for 1 epoch and observed improvements across all tasks, however we likely need to train additional epochs for better convergence and potentially explore different architectures that are better suited for multi-temporal disaster monitoring tasks.
> | Method | Event Classification | Temporal Grounding | Location Grounding |
> |--------|---------------------|-------------------|-------------------|
> | VideoLlava | 49.72% | 11.11% | 17.25% |
> | TEOchat | 48.88% | 15.15% | 15.50% |
> | Gemini 2.0-flash | 50.07% | 18.02% | 13.74% |
> | GeoChat | 28.18% | 26.5% | **76.80%** |
> | GPT 4.1 | 39.12% | 21.43% | 21.63% |
> | Gemini Pro 2.0 | 72.06% | 14.01% | 33.81% |
> | TEO-chat finetuned (1/5 MONITRS-QA) | 88.69% | 70.72% | 23.25% |
> | TEO-chat finetuned (full MONITRS-QA) | **91.66%** | **76.05%** | 31.34% |
>
> Please let us know if we can clarify anything further, we are happy to do so.
>
> [1] Kuckreja, K., Danish, M.S., Naseer, M., Das, A., Khan, S. and Khan, F.S., 2024. Geochat: Grounded large vision-language model for remote sensing. In Proceedings of the IEEE/CVF Conference on Computer Vision and Pattern Recognition (pp. 27831-27840).
>
>
> [2] Schottlander, D., & Shekel, T. (2025, April 8). Geospatial Reasoning: Unlocking insights with generative AI and multiple foundation models. Google Research Blog.

---

### Official Review · Reviewer_GACM · 2025-06-29

**Rating:** 4
**Confidence:** 3

**Summary:**

The paper introduces MONITRS, a novel multimodal dataset comprising over 10,000 FEMA-documented disaster events. This dataset integrates temporal satellite imagery, natural language annotations derived from news articles, geotagged locations, and question-answer pairs. The authors leverage MONITRS to fine-tune multimodal large language models (MLLMs) for disaster monitoring tasks, achieving significant performance improvements in event classification (88.69% vs. ~50% for baselines) and temporal grounding (70.72% vs. 11-18%).

**Dataset Code Accessibility:**

Partly

**Ethical Considerations:**

No, there are no or only very minor ethics concerns

**Final Justification:**

The authors have adequately addressed most of my concerns, and I find the manuscript acceptable for NeuRIPS.

**Limitations Weaknesses:**

1. Benchmarking Limitations:
The paper compares its model against VideoLLaVA, TEOChat, and Gemini 2.0-flash but omits comparisons with other prominent multimodal LLMs, such as GPT-4V. This gap limits the ability to fully contextualize the model’s performance against the current state-of-the-art in multimodal learning.
2. Location Grounding Challenges:
The model achieves only 23.26% accuracy in location grounding (vs. 13-17% for baselines), indicating a persistent challenge. While statistically significant improvements are noted (p < 0.01), the authors acknowledge that additional data (e.g., location embeddings, segmentation masks) may be required, suggesting an area for future work.
3. Experimental Details in the Paper:
Although implementation details are promised in the code release, the paper itself could benefit from including more specifics (e.g., hyperparameters, training procedures) to enhance immediate understanding without relying on external resources.

**Strengths Contributions:**

1. Dataset Contribution:
MONITRS is a pioneering dataset that combines satellite imagery with natural language annotations, covering a diverse range of disaster types (e.g., hurricanes, severe storms) and their full lifecycle (from onset to recovery).
With nearly 10,000 events and an average of 4.15 images per event spanning 18.14 days, it provides unprecedented temporal granularity and scale, making it a valuable resource for the field.
2. Reproducibility:
The authors provide open access to the dataset and commit to releasing the code upon acceptance, aligning with NeurIPS guidelines.
Implementation details and compute resource information are included, facilitating replication of the experiments.

---

> ### Author Rebuttal · Authors · 2025-07-31
>
> Thank you for your thorough review and constructive feedback. We hope to address your concerns below.
>
> **(1)** We agree that broader model comparisons strengthen our evaluation. During this rebuttal period, we have expanded our evaluation to include additional prominent MLLMs (Gemini 2.0 Pro, GPT-4.1). We also evaluate GeoChat [2], a single image remote sensing vlm, (using only a single image from the original sequence).
>
> We see a performance discrepancy between Gemini-Pro (72.06% event classification) and GPT-4.1 (39.12% event classification) with Gemini-Pro substantially outperforming GPT-4.1 on event classification tasks. We hypothesize that it is likely that Google has trained Gemini with labeled satellite imagery [1]. For temporal grounding, both models show modest performance (Gemini-Pro: 14.01%, GPT-4.1: 21.43%), while for location grounding, Gemini-Pro achieves 33.81% compared to GPT-4.1's 21.63%. This demonstrates that we have effectively created a challenging enough benchmark that even prominent MLLMs have significant room for improvement.
>
> **(2)** Yes, the location positioning task is difficult for some models, but **our new findings support that this is a valid task that sufficiently trained models should be able to perform.** Notably, GeoChat achieves exceptional performance on location grounding (76.80%, which supports our hypothesis that models specifically trained on geospatial relationships can excel at spatial localization tasks.
>
> To clarify the task is the following: We give the models the center coordinates of the image as well as the pixel resolution, and ask it to deduce the location of a concept/feature within the image in pixel coordinates. The understanding of pixel correspondence to latitude and longitude is non trivial, as the distance covered by 1 unit longitude or latitude is different at different locations around the globe.
>
> In the future as models utilize location encodings or additional modalities of information that encode geospatial coordinates, we should see improved performance in this task.
> We see far improved results with Geochat likely due to the fact that this model had a significant portion of its training data relating to the relationship between latitude and longitude and pixel correspondence [5]. We also find that multiframe or models that accept sequences of images actually perform worse than single image models like GeoChat for tasks such as location grounding. However, this multi-frame architecture is still necessary to classify or understand the progression of temporal events, as demonstrated by the event classification or temporal grounding results.
>
> | Method | Event Classification | Temporal Grounding | Location Grounding |
> |--------|---------------------|-------------------|-------------------|
> | VideoLlava | 49.72% | 11.11% | 17.25% |
> | TEOchat | 48.88% | 15.15% | 15.50% |
> | Gemini 2.0-flash | 50.07% | 18.02% | 13.74% |
> | GeoChat | 28.18% | 26.5% | **76.80%** |
> | GPT 4.1 | 39.12% | 21.43% | 21.63% |
> | Gemini Pro 2.0 | 72.06% | 14.01% | 33.81% |
> | TEO-chat finetuned (1/5 MONITRS-QA) | **88.69%** | **70.72%** | 23.25% |
>
> **(3)** Thank you for highlighting the need for more experimental details. We provide the complete training configuration below:
> We fine-tune TEOChat with the following experimental settings: LoRA fine-tuning (r=32, α=64), learning rate 2e-5, batch size 1 with gradient accumulation steps 8, cosine scheduler with 0.03 warmup ratio, and 8-bit quantization. Training used 4 A6000 GPUs for 1 epoch.
> We will revise the paper and code to include the more comprehensive implementation details in the final revision as well as the complete evaluations with the new experiments included.
>
> Please let us know if there are any more details that would help clarify anything - we are more than happy to discuss further.
>
> [1] Schottlander, D., & Shekel, T. (2025, April 8). Geospatial Reasoning: Unlocking insights with generative AI and multiple foundation models. Google Research Blog.
>
>
> [2] Kuckreja, K., Danish, M.S., Naseer, M., Das, A., Khan, S. and Khan, F.S., 2024. Geochat: Grounded large vision-language model for remote sensing. In Proceedings of the IEEE/CVF Conference on Computer Vision and Pattern Recognition (pp. 27831-27840).

---

### Official Review · Reviewer_ajrD · 2025-07-02

**Rating:** 5
**Confidence:** 3

**Summary:**

This paper introduces MONITRS, a novel, large-scale dataset for natural disaster monitoring. It pairs temporal satellite imagery for approximately 10,000 U.S. disaster events with multi-modal annotations, including natural language descriptions and question-answer pairs derived from news articles. Apart from the dataset itself, the contributions include a data-curation pipeline that uses LLMs to process FEMA records and news articles to generate temporally-aware annotations. The authors demonstrate its utility by fine-tuning an existing model on MONITRS, achieving significant performance gains on disaster-related VQA tasks.

**Dataset Code Accessibility:**

Yes

**Dataset Code Comments:**

No comments needed. The documentation within the paper is thorough and provides a clear methodology for how the data was sourced, filtered, and processed. This detailed description supports the accessibility and potential for future extensions of the work.

**Ethical Considerations:**

No, there are no or only very minor ethics concerns

**Limitations Weaknesses:**

The dataset's reliance on FEMA records limits its geographic scope to the United States, as the authors acknowledge in the limitations section. It may affect the generalizability of models trained on MONITRS to disasters in other parts of the world with different visual and ecological characteristics.

The proposed annotation pipeline depends on an LLM to generate captions and QA pairs from news articles, which may be a potential source of weakness. There is a risk of factual drift where the generated text may not perfectly correspond to the visual evidence in the satellite imagery. The authors may discuss any quality control or human-in-the-loop verification steps used to validate the LLM-generated annotations.

**Strengths Contributions:**

The work addresses a well-motivated gap: the lack of large-scale, temporally rich, and linguistically annotated datasets for general-purpose disaster monitoring. The proposed data collection pipeline is a novel and valuable contribution, effectively leveraging public data sources (FEMA, news articles) to create a resource of impressive scale and detail. The paper provides empirical evidence for the dataset's value. By establishing a new benchmark (MONITRS-QA) and showing that fine-tuning dramatically improves a model's ability to perform complex reasoning tasks like event classification (from ~50% to 88.7%) and temporal grounding (from ~15% to 70.7%), the authors clearly demonstrate the utility of their contribution. The presentation is clear, with informative figures and well-structured experiments.

---

> ### Author Rebuttal · Authors · 2025-07-31
>
> Thank you for your positive assessment and feedback! We hope to address your concerns below:
>
> **(1)** While we acknowledge the geographic limitation to US disasters, discussed in section 7 (page 9, lines 277-278), restricting to FEMA records allows us to at least partially confirm the locations and timing of these events against an existing database. Global datasets for geocoded natural disasters such as GDIS (geocoded disasters from NASA) [1] or the EM-DAT[2] are  geocoded at the country/province/regional level, which is much coarser than FEMA, making it difficult to acquire the precise satellite imagery required. To our knowledge, no similar scale, validated set of global geolocated natural disasters exists in an open source/ publicly available format. As such, our goal was to create a benchmark with the available FEMA data  so that the community can start working on this problem. That said, concerns about generalization are valid.  We will create a much smaller handcrafted dataset for global disasters and revise the paper to include this evaluation.
>
> **(2)** With regard to potential bias in LLM-generated annotations, we agree that LLMs are susceptible to biases in the input text, which is why we do not use a single article to create captions. We mitigate this potential bias and reduce noise by utilizing the information from multiple (5) relevant articles(section 3.1, page 5, lines 142-143) so that our captions are not biased by a single news article. However, we agree that some amount of human-in-the-loop quality control is important and will include such a study in our paper.
>
> [1] Rosvold, E.L., Buhaug, H. GDIS, a global dataset of geocoded disaster locations. Sci Data 8, 61 (2021).
>
> [2] Guha-Sapir, Debarati, Below, Regina, & Hoyois, Philippe (2014). EM-DAT: International disaster database. Centre for Research on the Epidemiology of Disasters (CRED).

---

### Official Review · Reviewer_4MH9 · 2025-07-20

**Rating:** 5
**Confidence:** 4

**Summary:**

This paper presents a multimodal dataset named MONITRS, designed for monitoring natural disasters through remote sensing technology. The dataset integrates approximately 10,000 natural disaster events recorded by the Federal Emergency Management Agency (FEMA) of the United States, encompassing time-series satellite imagery (Sentinel-2 data), natural language annotations derived from news articles, geographically tagged locations, and question-answer pairs. It aims to address issues present in existing disaster monitoring datasets, such as "focusing on a single disaster type," "insufficient temporal granularity," and "lack of natural language annotations." Experiments conducted on this dataset by fine-tuning multimodal large models (such as TEOchat) demonstrate significant performance improvements in tasks such as disaster classification and temporal localization, providing a new benchmark for machine learning-assisted disaster response systems.

**Dataset Code Accessibility:**

Yes

**Ethical Considerations:**

No, there are no or only very minor ethics concerns

**Final Justification:**

The restriction issue I was concerned about has been effectively resolved, and I have revised the score.

**Limitations Weaknesses:**

Geographic limitations: only includes disaster events within the United States (relying on FEMA data), lacking global diversity, which may limit the model's generalization ability in other regions (such as developing countries).
Sensor and resolution limitations: Dependent on Sentinel-2's RGB band (10m resolution, 5-day revisit period), it is difficult to capture rapidly changing details (such as immediate damage after an earthquake), and lacks all-weather imaging data such as SAR (ineffective for cloud coverage areas).
The performance of location positioning task is weak: the position positioning accuracy of the fine-tuning model is only 23.25% (Table 2), indicating that existing methods are difficult to accurately identify specific locations from images (such as the disaster situation of a certain road), and need to be combined with finer grained geographic information (such as map vector data).
Potential bias in LLM generated annotations: Natural language descriptions generated by Gemini may be influenced by subjective biases in news texts (such as exaggerated expressions in media reports), resulting in annotations that are inconsistent with the actual content of the image.
The experimental scale is limited: fine-tuning only used 1/5 of the training data and was not compared with more remote sensing specific models (such as GeoChat), which may underestimate the potential of the dataset.

**Strengths Contributions:**

Innovation
Problem oriented novelty
Directly addressing the core pain point of existing disaster monitoring: Existing remote sensing datasets mostly focus on static land use classification, lack fine-grained characterization of the "temporal evolution" of natural disasters, and rely on manual interpretation by experts. MONITRS combines FEMA disaster records with news texts for the first time, focusing on "dynamic monitoring of the entire lifecycle of disasters (from occurrence to recovery)", filling the gap in cross modal alignment of "temporal remote sensing data+natural language description".
The uniqueness of data construction
Multi source fusion method: Extracting the precise geographical location of disasters (such as specific roads and towns) through news articles, which compensates for the roughness of FEMA only recording county level coordinates (as shown in Figure 2 for accurate positioning of fire cross county distribution).
Temporal integrity: Satellite imagery covers the entire evolution process from 10 days before the occurrence of a disaster to 10 days after its end, solving the problem of limited time windows in existing datasets.
Cross modal annotation: Using news text to generate natural language descriptions of disaster time nodes (such as the daily records of wildfire spread in Figure 3), achieving triple alignment of "image text time".
Innovation in task design
The paper constructed the MONITRS-QA question and answer dataset, which includes tasks such as "event classification", "time localization", and "location localization" (Table 1). It not only evaluates the model's understanding of static images, but also emphasizes its ability to reason about "temporal changes" (such as determining the time of the first occurrence of disasters), promoting the paradigm upgrade of disaster monitoring from "static recognition" to "dynamic reasoning".
Advantages
Data scale and diversity: covering 9996 disaster events, including multiple types such as hurricanes, wildfires, floods, etc. (Figure 4), with a wide geographical distribution, avoiding the limitations of existing datasets with a single disaster type.
The reusability of technical methods: The proposed construction pipeline of "FEMA records+news text+satellite images" can be extended to other fields that require fine-grained localization and cross modal annotation, such as public health event monitoring.
The empirical effect is significant: the fine tuned model has significantly improved performance in event classification (88.69% vs baseline 50%) and time localization (70.72% vs baseline 18%) (Table 2), and the significance has been verified through statistical testing (McNemar's test, p<0.001), proving the improvement effect of the dataset on the model's ability.

---

> ### Author Rebuttal · Authors · 2025-07-31
>
> Thank you for your review, we appreciate the thorough assessment and the recognition of our work's key innovations. We hope to address your concerns below:
>
> **(1)** While we acknowledge the geographic limitation to US disasters, discussed in section 7 (page 9, lines 277-278), restricting to FEMA records allows us to at least partially confirm the locations and timing of these events against an existing database. Global datasets for geocoded natural disasters such as GDIS (geocoded disasters from NASA) [1] or the EM-DAT [2] are  geocoded at the country/province/regional level, which is much coarser than FEMA, making it difficult to acquire the precise satellite imagery required. To our knowledge, no similar scale, validated set of global geolocated natural disasters exists in an open source/ publicly available format. As such, our goal was to create a benchmark with the available FEMA data  so that the community can start working on this problem. That said, concerns about generalization are valid.  We will create a much smaller handcrafted dataset for global disasters and evaluate generalization in time for a camera ready revision.
>
> **(2)** Sentinel-2 imagery is the highest temporal and spatial resolution satellite imagery that is publicly available, and so it is what we chose for this current dataset. We will release the complete metadata of the dataset including locations, bounding boxes and time frames so that researchers with access to higher resolution proprietary data can expand the dataset to additional resolutions using other sources.
>
> **(4)** With regard to potential bias in LLM-generated annotations, we agree that LLMs are susceptible to biases in the input text, which is why we do not use a single article to create captions. We mitigate this potential bias and reduce noise by utilizing the information from multiple (5) relevant articles (section 3.1, page 5, lines 142-143) so that our captions are not biased by a single news article.
>
> **(3 & 5)** On the topic of remote sensing specific models such as Geochat, as we discuss in Related Works (pg 3, section 2.2 lines 83-85 and 90-92), we specifically target temporal understanding of disaster events. This requires models capable of processing sequences of images. GeoChat is designed for single-image analysis and cannot accept temporal sequences. TEOChat [3], which we do evaluate, was actually built upon GeoChat's foundation but extends it for temporal reasoning with satellite imagery sequences which made it the appropriate baseline for our temporal disaster monitoring tasks. However, this being said, we agree that including results with remote sensing specific models such as Geochat would strengthen our claims!Our goal was to create a challenging benchmark for multitemporal events in the satellite imagery domain and we believe our new findings support this further, so thank you for bringing this to our attention.
>
> We further evaluated our full dataset using prominent MLLMs (Gemini 2.0 Pro and ChatGPT 4.1) as well as GeoChat (using only a single image in the sequence) and include the results below.
>
> We agree with the reviewer that the location positioning task is difficult for some models, but **our new findings support that this is a valid task that sufficiently trained models should be able to perform.** Notably, GeoChat achieves exceptional performance on location grounding (76.80%), which supports our hypothesis that models specifically trained on geospatial relationships can excel at spatial localization tasks.
>
> To clarify the task is the following: We give the models the center coordinates of the image as well as the pixel resolution, and ask it to deduce the location of a concept/feature within the image in pixel coordinates. The understanding of pixel correspondence to latitude and longitude is non trivial, as the distance covered by 1 unit longitude or latitude is different at different locations around the globe. In the future as models utilize location encodings or additional modalities of information that encode geospatial coordinates, we should see improved performance in this task.
>
> We see far improved results with Geochat, likely because **this model had a significant portion of its training data relating to the relationship between latitude and longitude and pixel correspondence [5].** We found that multiframe or models that accept sequences of images actually perform worse than single image models like GeoChat for tasks such as location grounding. However, this multi-frame architecture is still necessary to classify or understand the progression of temporal events.
>
> We also see a performance discrepancy between Gemini-Pro and GPT-4.1 with Gemini-Pro substantially outperforming GPT-4.1 on event classification tasks. We hypothesize that it is likely that Google has trained Gemini with labeled satellite imagery [4]. This demonstrates that we have effectively created a challenging enough benchmark that even prominent MLLMs have significant room for improvement.
>
> **(5)** For the scale of experimentation, we agree that a full-dataset training experiment would further strengthen our claims. For the rebuttal, we have conducted an initial training with the full dataset for 1 epoch and observed improvements across all tasks, however we likely need to train additional epochs for better convergence and potentially explore different architectures that are better suited for multi-temporal disaster monitoring tasks.
>
> | Method | Event Classification | Temporal Grounding | Location Grounding |
> |--------|---------------------|-------------------|-------------------|
> | VideoLlava | 49.72% | 11.11% | 17.25% |
> | TEOchat | 48.88% | 15.15% | 15.50% |
> | Gemini 2.0-flash | 50.07% | 18.02% | 13.74% |
> | GeoChat | 28.18% | 26.5% | **76.80%** |
> | GPT 4.1 | 39.12% | 21.43% | 21.63% |
> | Gemini Pro 2.0 | 72.06% | 14.01% | 33.81% |
> | TEO-chat finetuned (1/5 MONITRS-QA) | 88.69% | 70.72% | 23.25% |
> | TEO-chat finetuned (full MONITRS-QA) | **91.66%** | **76.05%** | 31.34% |
>
> We will revise the paper to include the results with the full set of new experiments. Please let us know if we can clarify anything further, we are happy to do so.
>
> [1] Rosvold, E.L., Buhaug, H. GDIS, a global dataset of geocoded disaster locations. Sci Data 8, 61 (2021).
>
> [2] Guha-Sapir, Debarati, Below, Regina, & Hoyois, Philippe (2014). EM-DAT: International disaster database. Centre for Research on the Epidemiology of Disasters (CRED).
>
> [3] Irvin, J.A., Liu, E.R., Chen, J.C., Dormoy, I., Kim, J., Khanna, S., Zheng, Z. and Ermon, S., ICLR 2024. Teochat: A large vision-language assistant for temporal earth observation data. arXiv preprint arXiv:2410.06234.
>
> [4] Schottlander, D., & Shekel, T. (2025, April 8). Geospatial Reasoning: Unlocking insights with generative AI and multiple foundation models. Google Research Blog.
>
> [5] Kuckreja, K., Danish, M.S., Naseer, M., Das, A., Khan, S. and Khan, F.S., 2024. Geochat: Grounded large vision-language model for remote sensing. In Proceedings of the IEEE/CVF Conference on Computer Vision and Pattern Recognition (pp. 27831-27840).

---

> > ### Comment · Reviewer_4MH9 · 2025-08-05
> >
> > Thank you for the author's response. The restriction issue I was concerned about has been effectively resolved, and I have revised the score.

---

> > > ### Author Response · Authors · 2025-08-05
> > >
> > > Thank you, we are glad we were able to address your concerns.

---

### Note · Authors · 2025-08-15

Thank you to the reviewers for their thoughtful engagement with our dataset work. We were pleased to see the appreciative reception and were able to address raised concerns in our rebuttal. The positive feedback from reviewers 4MH9 and ajrD was particularly encouraging, with reviewer 4MH9 confirming that our rebuttal response "effectively resolved" their concerns and indicating their intention to revise their score upward. We will revise the paper to incorporate the clarifications, additional results, and expanded discussions from our rebuttal.

---

### Decision · Program_Chairs · 2025-09-18

**Decision:**

Accept (spotlight)

**Comment:**

**Summary (1–2 lines):** MONITRS pairs \~10k FEMA disasters with multi-temporal Sentinel-2 imagery, geo metadata, and news-derived captions/QA to benchmark event classification, temporal localization, and spatial grounding; fine-tuning on MONITRS yields large gains over generic MLLMs.

**Strengths**

* Clear gap filled: multi-temporal, language-aligned remote sensing for disasters (end-to-end lifecycle).
* Solid scale & breadth (≈10k events; diverse hazards) with a transparent curation pipeline.
* Useful tasks/metrics: baseline suite now expanded (VideoLLaVA, TEOChat, Gemini, GPT-4.1, GeoChat).
* Strong empirical signal: fine-tuned TEOChat jumps to \~92% event cls and \~76% temporal grounding (1-epoch full-data results reported).

**Weaknesses / risks (manageable)**

* U.S.-only (FEMA) focus; generalization beyond the U.S. not yet demonstrated.
* LLM-generated text may drift from imagery; limited human QA so far.
* Spatial grounding remains challenging for most models (GeoChat is a notable exception).
* Packaging: Croissant bot flags **missing license** on HF and one inaccessible file; RAI metadata is empty.

**Rebuttal takeaways** Authors added evaluations (Gemini Pro, GPT-4.1, GeoChat—very strong on location grounding), reported full-dataset fine-tuning (1 epoch), clarified the task spec for location grounding, and committed to (i) a small global set to probe transfer and (ii) releasing full geo/time metadata to enable higher-res/SAR extensions.

**Decision & rationale:** **Accept.** The dataset is timely, well-constructed, and already demonstrates scientific utility; reviewers converge on accept (two 5s, two 4s). Remaining issues are editorial/packaging, not fatal.

**Presentation recommendation:** **Poster (strong).** High interest for EO/ML \+ disaster response; spotlight could be considered if packaging fixes (license/RAI) land quickly.

**Actionable notes to authors (camera-ready)**

* Add an explicit **license** on HF (e.g., CC BY-4.0 to match paper) and fix the inaccessible file; populate Croissant **RAI fields**.
* Brief **human QA** pass on a sampled set of captions/QA with error taxonomy; document inter-annotator checks.
* Include **hyperparameters & training details** in the paper and repository; pin dependencies.
* Report a **global transfer** sanity check (your proposed small set).
* For spatial grounding: consider lat/long encodings, map vectors/roads, DEM/SAR options; provide a geodesy helper to convert pixels↔geocoords.
* Keep GeoChat as a **single-image spatial baseline**; retain multi-frame models for temporal tasks; add ablations on window length.

===== FINAL UPDATE FROM DB Track PCs ====

The final decision for this paper has been taken by the program chairs after consultation with the SACs. All Senior Area Chairs have ranked papers according to the feedback from the AC during the review process. We decided to leave the original meta-review to reflect the opinion of the AC in light of the initial discussions with reviewers and SAC.